# Deep Learning Approach to UAV Detection and Classification by Using Compressively Sensed RF Signal

**DOI:** 10.3390/s22083072

**Published:** 2022-04-16

**Authors:** Yongguang Mo, Jianjun Huang, Gongbin Qian

**Affiliations:** 1Guangdong Key Laboratory of Intelligent Information Precessing, College of Electronic and Information Engineering, ATR Key Laboratory, Shenzhen University, Shenzhen 518060, China; 2060432070@email.szu.edu.cn; 2College of Electronic and Information Engineering, Shenzhen University, Shenzhen 518060, China; qiangb@szu.edu.cn

**Keywords:** unmanned aerial vehicles, detection and identification, radio frequency, compressed sensing, deep learning

## Abstract

Recently, the frequent occurrence of the misuse and intrusion of UAVs has made it a research challenge to identify and detect them effectively, and relatively high bandwidth and pressure on data transmission and real-time processing exist when sampling UAV communication signals using the RF detection method. In this paper, firstly, for data sampling, we chose a compressed sensing technique to replace the traditional sampling theorem and used a multi-channel random demodulator to sample the signal; secondly, for the detection and identification of the presence, type, and flight pattern of UAVs, a multi-stage deep learning-based UAV identification and detection method was proposed by exploiting the difference in communication signals between UAVs and controllers under different circumstances. The data samples are first passed by detectors that detect the presence of UAVs, then classifiers are used to identify the type of UAVs, and finally flight patterns are judged by the corresponding classifiers, for which two neural network structures (DNN and CNN) are constructed by deep learning algorithms and evaluated and validated by a 10-fold cross-validation method, with the DNN network used for detectors and the CNN network for subsequent type and flying mode classification. The experimental results demonstrate, first, the effectiveness of using compressed sensing for sampling the communication signals of UAVs and controllers; and second, the detecting method with multi-stage DL detects higher efficiency and accuracy compared with existing detecting methods, detecting the presence, type, and flight model of UAVs with an accuracy of over 99%.

## 1. Introduction

Drones, also known as unmanned aerial vehicles (UAVs), are unmanned aircraft operated by radio-controlled equipment and self-contained programmable control devices or operated completely or intermittently and autonomously by an onboard computer. In recent years, the rapid development of the UAV industries has been driven by the maturing technologies in the fields of information, control, and communication, which has led to the gradual lowering of the threshold for the use of consumer- and industrial-grade UAVs, making them widely used in various civil and commercial applications, such as cinematography, agricultural inspection, and rescue search [1]. However, due to the lack of unified industry standards and norms, the threshold of access is low and most UAVs are operated in a “black flight” state. While bringing convenience to various industries, once there is abuse and illegal intrusion, it can seriously threaten major public events, important areas, airspace security, personal privacy, and national security, and even endanger human lives and cause property damage. For example, they can endanger the safety of important targets and affect the take-off and landing of passenger planes [2]; in addition, they may be used by lawless elements to spy on people’s privacy, smuggle drugs, or even become “suicide drones” by terrorists with bombs [3,4]. Though the government has performed a lot of work on drone regulation in recent years, including registration of real names, building a comprehensive monitoring platform for drone cloud systems, and setting up electronic fences, there are still a large number of hard-to-detect drones that need to be detected and identified by relevant monitoring technologies [5].

Given various problems caused by misuse and malicious intrusion of UAVs, the accurate detection and classification of UAVs are particularly important to ensure public and national security. Many experts and scholars have proposed many different UAV detection methods, which can be roughly divided into UAV detection algorithms based on acoustics [6,7], visual (image or video) [8,9], radar [10,11,12,13], and radio frequency [14,15]. Considering the first three methods in practical application, there are some limitations and deficiencies, and it is difficult to achieve a good detection effect. For example, the acoustic-based UAV detection method has such problems, such as susceptibility to interference from environmental or atmospheric noise and small detection range; the visual-based UAV detection method cannot be used in dim and severely obstructed vision scenarios; and the poor detection performance of the radar detection method is due to the low cross-sectional area of the UAV radar, which is easily obscured by obstacles such as walls and buildings [6,10]. In contrast, the RF fingerprint-based UAV detection method technique is based on listening to UAV communication signals by using the high-gain receiving antenna and high-sensitivity receiving system, which can solve the distance problems related to visual and acoustic techniques and the influence of relevant environmental factors, and the RF fingerprint detection method only requires a sufficient UAV dataset for training, which makes the RF detection method a promising solution.

UAVs are usually flying in the air to achieve a specific task. However, within the no-fly zone, there are techniques that need to be used to detect any drone intrusion in the area for security reasons. In this paper, firstly, for data acquisition, we choose to use compressive sampling [16,17] instead of the traditional sampling theorem since the UAV communication signal is sparse in nature, and for this reason, we construct a multi-channel random demodulation (MCRD) [18] sampling structure to sample and compress the UAV communication signal simultaneously. This reduces the signal bandwidth to some extent and relieves the pressure on data storage, transmission, and processing. Deep learning (DL) is a very popular field of machine learning in recent years, and unlike other machine learning algorithms, deep learning techniques are considered to be one of the most powerful and effective techniques for selecting, extracting, and analyzing features from the original dataset with relatively strong generalization capabilities. Therefore, we choose to use the deep learning approach to build neural networks for training and validation of the original UAV dataset.

In summary, to effectively and accurately detect and classify UAVs, we propose a UAV radio frequency fingerprint detection and classification algorithm based on compressed sensing and deep learning algorithms. The main contributions of this article are as follows:According to the characteristics of the UAV communication RF signal, the compressed sensing technology is introduced, and its effectiveness is verified through experiments.A deep learning UAV detection and classification network based on radio frequency compressed signals is constructed by using deep learning algorithms;Filtering and feature extraction are performed on the compressed measurement signal, which improves the classification effect of UAV types and modes.

The rest of this paper is organized as follows: Section 2 mainly summarizes the related work and techniques; Section 3 focuses on the proposed sampling, pre-processing, and detecting methods; Section 4 describes the dataset, experiments, evaluation metrics, and experimental results used in this paper; Section 5 compares our method with other detecting methods using the same dataset; and Section 6 concludes and provides an outlook on future work.

## 2. Related Work

In this section, we mainly introduce some work related to the RF fingerprint-based UAV detecting method. In general, the RF-based UAV detecting method has two main parts: one is to collect the background RF signals, the RF signals of various types of UAVs, and build the raw dataset available for training; the second is to train and validate the original dataset of drones using the corresponding detection and classification methods to finish the detection and classification of UAVs.

### 2.1. Data Acquisition

For data acquisition of UAV RF signals, many papers nowadays basically use the traditional Nyquist sampling theorem for sampling as they do not take into account the sparse nature of UAV communication signals. In [19], Al-Sa’d et al. address the lack of an open-source dataset of radio frequency (RF) signals for UAV communication by using two NI USRP-2943R RF receivers to collect, analyze, and record raw RF signals from UAVs in different flight modes at a bandwidth of at least 80 MHz by the sampling theorem to construct a UAV RF database. The sampling theorem requires that the sampling rate of the signal must be greater than or equal to twice the maximum frequency to recover the original signal from the sampled signal. This means that it will have a sampling rate of 160 MHz or more, which is fully manageable by the sampling system, but there is still a large pressure on the storage, transmission, and processing of data for a UAV detection system with high requirements for lightweight and real-time [20]. However, we note that the RF signals communicated between the UAV and the controller are sparse, which means that it is perfectly possible to use compressed sensing techniques to sample them instead of the Nyquist sampling theorem. Moreover, introducing the CS framework to UAV signal acquisition has at least several advantages: (i) the number of samples collected from sensors is significantly reduced, (ii) the power consumption of the CS-based acquisition system is decreased, (iii) the processing time required by the algorithms required for refinement is reduced, and (iv) the observation bandwidth of the overall acquisition system is increased [21].

Compressed sensing, also known as compressive sampling [16], breaks the limits of the traditional sampling theorem so that the sampling rate of the signal is no longer limited by the signal bandwidth, but depends on the content of the information in the signal. It takes advantage of the sparsity of the signal to obtain the compressed measure of the signal at a sampling rate lower than the Nyquist rate, samples and compresses the signal at the same time, and then recovers the original signal by a reconstruction algorithm. In [21], for the problem of detecting, locating, and tracking mobile RF transmitters, Pasquale et al. detail several emerging areas of this new RF sensing system, and also propose and discuss a technical approach to use compressed sensing for RF sensing systems, and determine its feasibility. It is also pointed out that the use of CS techniques for RF sensing systems does not require a complete signal reconstruction from compressed measurements. In [22], to solve the problem of real-time UAV data return to the receiving station with limited bandwidth, Huang et al. utilize compressed sensing and matrix complementation techniques to achieve real-time transmission of UAV monitoring data, which greatly reduces the data return time while restoring the image almost identical to the original one. In [23], to correctly classify ground targets by UAVs (UAVs), Zhu et al. used the PCA algorithm to remove ground clutter and refined the micro-Doppler characteristics with compressed sensing to extract features, which improved the classification accuracy and achieved an overall accuracy of 92.5%. In [24], Gaigals et al., established the detection of UAV transmission signals by passive radar and compressed and sampled the UAV transmission signals by compressed sensing technique to investigate the effect of different antenna settings on the UAV detection quality. It is fully demonstrated that the compressed sensing technique is fully applicable to the RF signal sampling between the UAV and controller.

### 2.2. Detection and Classification Methods

Target detection methods have evolved and an increasing number of detection methods are proposed. Examples include some adaptive detectors proposed for the generalized likelihood ratio test (GLRT), the Rao test, the Wald test, or modifications to them [12]; the total Bregman divergence-based matrix information geometry (TBD-MIG) detector [11]; and the Signal Subspace Matching Detector (SSM) [13], etc. However, the RF signal targets for UAVs are currently mainly detected and classified by traditional machine learning or deep learning-based methods.

#### 2.2.1. Traditional Method Based

The traditional machine learning-based UAV detecting method is simply to detect and classify the primary UAV dataset using traditional machine learning methods, such as kNN, XGboost, MLP, SVM, etc., to detect and classify the dataset.

In [25], Azuma et al. transformed the original signal to the wavelet domain in the detection phase and used a Naive Bayes Method to detect drones; in the classification phase, a set of the statistical features was extracted from the energy transient signal, then the important features were selected by Neighborhood Component Analysis (NCA), and finally the selected features were given back to several machine learning algorithms for classification, where the use of the kNN classification method achieved an accuracy rate of 96.3%. They experimented in [26] to achieve 98% classification accuracy using KNN under signal interference from Wi-Fi and Bluetooth transmitters at 25 db. In [27], Medaiyese et al. designed a machine learning RF-based DDI system with three machine learning models developed by the XGBoost algorithm, and experimentally verified that the low-frequency spectrum of the captured RF signal in the communication between the UAV and its flight controller as the input feature vector already contains enough information; and finally, using only the low-frequency spectrum as the input feature vector, the XGBoost algorithm was used to detect and identify the presence of an UAV, three types of UAVs, and corresponding flying modes with an average accuracy of 99.9%, 90.73%, and 70.09%, respectively. In [28], Ibrahim et al. proposed an improved UAV detection and identification method using an integrated learning approach based on hierarchical concepts combined with preprocessing and feature extraction stages of RF data. The detection method makes full use of the KNN and XGBoost methods in machine learning and is evaluated in the publicly available dataset of [15], which shows that the method can detect whether a UAV is flying in the region and can directly identify the type of UAV and corresponding flying modes with an average accuracy of about 99%.

#### 2.2.2. Deep Learning Based

DL is a very popular field of machine learning in recent years, unlike traditional machine learning algorithms; firstly, it can select extracts as well as analyze features directly from the original dataset without relying on manual feature selection and extraction; and secondly, DL techniques can adapt to a variety of different datasets without affecting the classification performance, which means that its generalization ability is very powerful.

In [29], Al-Sa’d et al., for their dataset published in [19], designed three deep neural networks (DNN) for UAV presence, type, and flying modes detection and classification, respectively, which were validated by 10-fold cross-validation and evaluated using various metrics. The classification results show that the average accuracy of the DNN method in detecting the presence, type, and corresponding flying modes of UAVs reached 99.7%, 84.5%, and 46.8%, respectively, which fully demonstrated the feasibility of UAV detection and recognition through the dataset. In [30], Al-Emadi et al. similarly based on this dataset, tried to solve the UAV detection and classification problem by adding a feature extraction layer with a convolutional layer then a fully connected layer for detection and classification, which improved the accuracy of UAV detection and classification. In [31], Shi et al. built an RF UAV dataset including five types of UAV by recording two types of UAV with yunSDR software radio devices based on the dataset [19]. A 1D convolutional neural network was designed for detection and identification based on the characteristics of the sample data and CNN network. The final accuracy of detecting the presence of UAVs was 99.6%; the accuracy of classifying UAV types was 96.8% (6 types); and the identity rate of flying modes reached 94.4% (12 modes).

## 3. Proposed Sampling and Detection Method

In this section, we mainly introduce the sampling structure and data pre-processing method based on CS theory that we adapt according to the signal characteristics of the UAV RF signals; secondly, we introduce the neural network structure designed in this paper for detecting and classifying UAV RF fingerprints.

### 3.1. Detection and Classification Methods

#### 3.1.1. Compressive Sensing

In this paper, the signal we sample is the RF signal x(t) of the session between the UAV and the controller. Although the x(t) signal is susceptible to various kinds of noise during communication, leading to a small percentage of zero elements in x(t), many elements are close to zero. It is continuously collected when it is sampled, but the session between the UAV and the controller is intermittent, so at the moment when there is no command transmission, the sampled signal is only noise, making the difference between the values of the elements of the x(t) signal very large. However, after filtering the signal x(t), x(t) is exhibiting sparsity in the frequency band; and when x(t) has sparsity, it just meets the prerequisite condition of compressive sensing, so the sampling of x(t) by compressive sensing is available.

With the development of compressed sensing technology, researchers have proposed a variety of sampling structures based on CS theory, whose representative results mainly include Random Filter (RF) [32], Random Sampling (RS) [33], Random Demodulation (RD) [34], and Modulated Wideband Converter (MWC) [35]. According to the characteristics of the x(t) signal, we constructed a sampling structure of multichannel random demodulation (MCRD) based on CS theory, as shown in Figure 1, which is composed of multiple channels, each of which can be considered as an independent RD system. The structure is similar to MWC, but the randomly mixed signal is different. The mixing function of MWC is periodic, while the mixed signal of MCRD is a random sequence, so only a random sequence needs to be generated in MCRD, and then modulated into different carrier frequencies.

Suppose the UAV communication signal is x(t) with a bandwidth of B. It is divided into m subbands, each with a bandwidth of B/m. Then the signals of these m subbands are multiplied with the random sequence separately and then sampled at low speed (sampling frequency lower than 2 B/m) after passing low-pass filtering to obtain compressed sampling for multiband random demodulation. The specific steps are as follows: the x(t) signal enters m channels at the same time, and in i-th) channel.

x(t) is multiplied with the mixed signal $p_{i}\left(t\right)$ time domain, and the spectrum of x(t) is frequency shifted to the lower frequency band; that, is the spectrum of x(t) is moved to the [−f/m, f/m] region, thus the mixed signal is:(1)$$p_{i}\left(t\right)=2\varepsilon_{n}cos\left(2\pi \frac{i\cdot f}{m}t\right),\;\;\;t\in \left[\frac{n}{W},\frac{n+1}{W}\right),\;n=0,1,\cdots W-1,\;i=0,1,\cdots ,m-1,$$
where $\varepsilon_{n}$ is obtained with equal probability taking ± 1 and *W* is the Nyquist rate.After the ideal low pass filter h(t) with a cut off frequency of $f_{s}/2$,the mixed output signal ${\tilde{x}}_{i}\left(t\right)$ will be filtered out of the high frequency signal and only the signal whose spectrum is shifted to the low frequency band (the narrow band signal with frequency in [−/2, $f_{s}$/2]) will be retained, so the filtered signal is the signal of the i-th band in the original x(t).Finally, the m group of low-speed sampling sequence $y_{i}\left(n\right)$ is obtained by sampling interval $T_{s}$ for the low-speed ADC sampling (ADC sampling satisfies the Nyquist theorem), when:(2)$$y_{i}\left(n\right)=\int_{0}^{n\cdot T_{s}}x\left(\tau \right)d\tau \cdot \int_{0}^{n\cdot T_{s}}p_{i}\left(\tau \right)h\left(n\cdot T_{s}-\tau \right)d\tau =\Phi_{i}\cdot x\left(n\right),$$
(3)$$\Phi_{i}=\int_{0}^{n\cdot T_{s}}p_{i}\left(\tau \right)h\left(n\cdot T_{s}-\tau \right)d\tau ,$$

In this paper, because the high-speed pseudo-random sequence (mixed signal $p_{i}\left(t\right)$ is produced by taking ±1 with equal probability, after low-pass filtering and low-speed sampling, the measurement matrix of the whole MCRD sampling structure becomes a random Bernoulli matrix.
(4)$$\Phi =\left[\begin{array}{ccc} \Phi_{0} & \cdots & 0\\ \vdots & \begin{array}{cc} \Phi_{1} & \\ & \ddots \end{array} & \vdots \\ 0 & \cdots & \Phi_{m-1} \end{array}\right],$$

If the Nyquist sample of signal ${\tilde{x}}_{i}\left(t\right)$ on a frequency band of the x(t) signal is ${\tilde{x}}_{i}\left(m\right)$ with length M, then the compressed measurement signal $y_{i}\left(n\right)$ in the corresponding ith channel in the MCRD is mathematically represented as the product of the observation matrix Φ ∈$\;R^{n\times m}$ and ${\tilde{x}}_{i}\left(m\right)$.
(5)$$y_{i}\left(n\right)=\Phi_{i}\times x_{i}\left(m\right),$$

#### 3.1.2. Data Pre-Processing

The RF signal x(t) communicated between the drone and the controller is sampled by the CS-based MCRD sampling structure to obtain the compressed measurement signal $y_{i}\left(n\right)$ for each channel; however, the data are still further processed to make it easier to distinguish the characteristics of the data and to collocate each $y_{i}\left(n\right)$. The specific preprocessing is as follows.

Zero-centered the compressed measurement signal $y_{i}\left(n\right)$, to remove mainly the zero-frequency component and the offset component.Computing the $y_{i}\left(n\right)$ power spectral density $P_{i}\left(k\right)$.
(6)$$P_{i}\left(k\right)=\frac{1}{N}\sum_{m=0}^{N-1}{\left[y_{i}\left(n\right)e^{j\frac{2\pi mk}{N}}\right]}^{2},$$
where *N* is the length of $y_{i}\left(n\right)$, and k≤n.The conversion signals of each channel are connected to create the complete spectrum. As the connection between each conversion signal obtained after MCRD sampling is the same. Therefore, only the connection of two adjacent channels is given here.
(7)$$P\left(k\right)=\left[P_{i}\left(k\right),\;cP_{i+1}\left(k\right)\right],$$
(8)$$c=\frac{\sum_{q=0}^{Q}P_{i}\left(k\right)\left(N-q\right)}{\sum_{q=0}^{Q}P_{i+1}\left(k\right)\left(q\right)}\;,q=0,1,\ldots ,Q-1$$
where c is a normalization factor calculated as the ratio between the last Q samples of the previous power spectrum $P_{i}\left(k\right),$ and the first Q samples of the next power spectrum $P_{i+1}\left(k\right)$, and N is the total number of in $P_{i}\left(k\right)$. c ensures that continuity of the spectrum is maintained between the two halves of the RF spectrum, as they are captured using different devices; however, the spectral bias is inevitable. Note that Q must be relatively small to successfully stitch the two spectra and large enough to average out any random fluctuations, e.g., Q = 10 for M = 2048.Finally, only the connected power spectral density P needs to perform maximum normalization to vary all the values to within the interval from 0 to 1, which constitutes one piece of data for the input detection classification network.

### 3.2. Two/Multistage Classification Network

In this paper, we directly feature extraction and identify the compressed sampled signal without recovering the original signal through reconstruction algorithms. For such signals, feature extraction by conventional methods is relatively difficult, but DNN and CNN networks extract features by learning methods; the latter obviously being a little easier. Secondly, many recent research results [29,30,31] have shown that DNN and CNN networks are perfectly suitable for detecting and classifying RF signals. Therefore, the DNN and CNN network structures are chosen for detection and identification. Below, the CNN and proposed network structure is described in detail.

CNN is a class of DL techniques that has the unique feature of combining feature extraction and classification into a single model. It is composed of layers, each layer is a collection of neurons, and consists of three main layers: convolutional layer, pooling layer, and fully connected layer. The convolutional layer mainly extracts features from the input data; the pooling layer is used to reduce the computational complexity and cost by downsampling; and finally, the fully connected layer and the corresponding activation function are used to classify the features extracted from the previous layer. Thus, CNN provides a flexible architecture to increase and decrease the number of layers according to the need and the complexity of the data.

In this paper, we need to detect and classify the RF signals of communication between the UAV and controller. Within the no-fly zone, we need to detect the presence of the UAV, and identify the type of UAV and the current flying modes of that type of UAV. For this case, we can use a layered learning approach, where the layered model first determines the presence of the UAV; then, identifies the type of UAV based on that RF signal after the presence of the intruding UAV is determined; finally, the flying modes in which the UAV is located are further determined based on that type of RF signal. Given the different detection and classification complexities, we designed the following two different neural network models for them, respectively, as follows.

**UAV Detection Network** (DNN Network): mainly used in the no-fly area to detect the existence of UAVs; in other words, two types, with and without the presence of UAVs. Since these two signals are still relatively easy to distinguish in the spectrum, we constructed a DNN network consisting of five Dense layers, with the specific parameters shown in Table 1.


**Table 1 sensors-22-03072-t001:** UAV Detection Network.

Layer	Embedded Structure	Parameter	Activation
1	Input	(None, 2047)	-
2	Dense Dense Dense Dense	256 128 256 128	ReLU ReLU ReLU ReLU
3	Output Layer	2	sigmoid

2.**UAV Identification and Classification Network** (CNN Network): mainly used to identify and classify the type and flying modes of UAVs. Firstly, after detecting an intruding UAV, the type of UAV (N class) is identified based on the RF signal, where N is the number of UAV type contained in the UAV dataset; secondly, after determining the type of UAV, then the motion mode under that type is further determined (4 Classes: on, connected and off; hovering; flying; and flying with video recording). For both cases, we use convolutional layers for feature extraction, pooling layers to reduce the information size, and finally fully connected layers for classification. A CNN network structure is designed, consisting of six 1D convolutional layers, each followed by a pooling layer, ending with two fully connected layers for further classification, interspersed with two Dropout layers to prevent overfitting, with the parameters shown in Table 2.


**Table 2 sensors-22-03072-t002:** UAV Identification and Classification Network.

Layer	Embedded Structure	Parameter	Activation
1	Input Layer	(None, 2047, 1)	-
2	Conv1D Max pooling	filters = 32, kernel size=6	ReLU
3, 4, 5	Conv1D Average pooling	filters = 64, kernel size = 3	ReLU
6, 7	Conv1D Average pooling	filters = 128, kernel size = 3	ReLU
8	Dropout Flatten	0.25	-
9	Dense Dropout Dense	256 0.22 128	ReLU
10	Output Layer	Type of Output	softmax

## 4. Experiments and Results

In this section, we first introduce the dataset used, followed by the detailed sampling process further in the simulation experiments, and finally the evaluation metrics and experimental results for performing the detection and classification.

### 4.1. Dataset

During the experiments, we choose the dataset publicly available in [19] to better compare with existing work. Based on this dataset, the compressed sampled dataset is simulated.

#### 4.1.1. Raw Dataset

The dataset contains 227 segments of recorded RF signal strength data, each segment includes two parts: low-frequency signal $x_{l}\left(n\right)$ and high-frequency signal $x_{h}\left(n\right)$; each part contains 1 million samples. They obtained from 3 UAVs (AR, Bebop, and Phantom) experiments with 10 types. There was 10.25 s of background RF activity data (no UAV), and 5.25 s of UAV communicated RF data. Besides the background RF activity, four different modes or states of the UAV are recorded: on, connected and off; hovering; flying; and flying with video recording, as shown in Table 3. The UAV is controlled by the controller, and the RF signal strength data are collected by two RF receivers that receive the first 40 MHZ (low-frequency signal) and the second 40 MHZ strength (high-frequency signal).

#### 4.1.2. Simulation Dataset

In the dataset of [19], it is known that the UAV communication signals x(n) are all already acquired, which are split into two low-frequency signals $x_{l}\left(n\right)$ and high-frequency signals $x_{h}\left(n\right)$ with temporal continuity and sparsity, so that the number of channels m in the MCRD sampling structure of Section 3.1 is two when the random Bernoulli matrix becomes:(9)$$\Phi =\left[\begin{array}{cc} \Phi_{1} & 0\\ 0 & \Phi_{2} \end{array}\right],$$

For increasing the number of samples in the dataset, we divide the $x_{l}\left(n\right)$ and $x_{h}\left(n\right)$ signals into 100 segments and record each segment as signal $x_{l/h}{}_{i}\left(n\right)$ with length $1\times 10^{5}$. Based on the length of the $x_{l/h}{}_{i}\left(n\right)$ signal and compressed measurement signal, we generated the 2048$\times 1*10^{5}$ random Bernoulli matrix $\Phi_{1/2}$. Then, the $x_{l/h}{}_{i}\left(n\right)$ and $\Phi_{1/2}$ are substituted into the Equation (5) and low-pass filtered to obtain the corresponding low-dimensional observation signal with length 2048. After obtaining the low-dimensional observed signal $y_{l/h}{}_{i}\left(m\right)$, only feature extraction and spectral concatenation need to be performed according to the preprocessing link in Section 3.1. Note: To reduce the amount of data and considering that the power spectral density $P_{i}\left(k\right)$ is symmetric here, it is sufficient to take half of each of the two channels $P_{i}\left(k\right)$ for spectral connection.

#### 4.1.3. Dataset Reliability Analysis

For preliminary validation of the method in this paper and to ensure the reliability of the dataset fed into the neural network, we statistically analyzed and initially examined the spectrum and correlation features of the RF signal dataset generated by simulated compressed sampling. After noise variation reduction and aggregation with smoothing of the spectrum, the average power spectra of the three given scenarios (Class 2, 4, and 10) in the dataset were obtained as shown in Figure 2. In two classes of scenarios, Class 1 represents the RF background activity in the absence of UAVs, and Class 2 represents the RF signal in the presence of UAVs. For four classes of scenarios, Class 1 represents RF background activity without UAV, Class 2 to 4 are Bebop, AR, and Phantom, respectively. For 10 classes, Class 1 represents RF background activity without UAV, Class 2 to 5 represents the four modes of Bebop, Class 6 to 9 represents the four modes of AR and Class 10 is Phantom.

As can be seen from Figure 2, there are certain differences between the average power spectra of different types of UAVs, and there are also certain differences between the average power spectra of different flight modes of the same type of UAV, but the differences are relatively small, so it is difficult to make accurate classification using traditional signal recognition methods; hence, we introduce convolutional neural networks to dig deeper into their intrinsic deep features for signal differentiation.

### 4.2. Counterpart Two/Multilevel Classification

For this dataset, our approach is shown in Figure 3. The raw RF signal from the drone is pre-processed with data to obtain the corresponding simulated data, which then enters the Detector in the learning phase. The output of this Detector on our RF signal informs whether there is a drone or no drone in the detected area. If there is a drone in the area, this data sample enters Classifier 0 to determine the type of drone detected (Bebop, AR, and Phantom 3). If the detected drone is Phantom 3, the output of the detection method is category number 9. Otherwise, the data sample enters Classifier 1, when the detected drone is Bebop, or Classifier 2, when the detected drone is AR. Classifiers 1 and 2 define the mode of the detected drone as one of the following based on four classes: on, connected and off; hovering; flying; and flying with video recording, where the classes for Bebop UAV are: 1 for on, connected and off mode; 2 for hover mode; 3 for flying mode; and 4 for flying with video recording mode. For the AR UAV, the modes are as follows: 5 for on, connected and off mode; 6 for hover mode; 7 for flying mode; and 8 for flying with video recording mode. The Detector uses the UAV detection network (DNN structure) from Table 1; the Classifier 0, 1, and 2 use the identification and classification network (CNN structure) from Table 2.

### 4.3. Detection and Classification

#### 4.3.1. Assessment Indicators

In our process of challenging UAV detection and classification systems, we have added recall, error rate, precision, false-negative rate (FNR), false discovery rate (FDR), and F1 score to assess the goodness of the system, taking into account the fact that classes are unequally correlated with each other and that classes are unbalanced, in addition to using accuracy to assess the goodness of the system. These metrics are assessed and summarized in matrix form using a confusion matrix, which can be calculated from the equation below.
(10)$$\mathrm{Accuracy}=\frac{\mathrm{TP}+\mathrm{TN}}{\mathrm{TP}+\mathrm{TN}+\mathrm{FP}+{\mathrm{FN}}^{\prime }}\;,$$
(11)$$\mathrm{Precision}=\frac{\mathrm{TP}}{\mathrm{TP}+{\mathrm{FP}}^{\prime }}\;,$$
(12)$$\mathrm{Recall}=\frac{\mathrm{TP}}{\mathrm{TP}+{\mathrm{FN}}^{\prime }}\;,$$
Error = 1 − Accuracy(13)
FDR = 1 − Precision(14)
FNR = 1 − Recall(15)
(16)$$F1\;\mathrm{score}=2\times (\frac{\mathrm{Precision}\times \mathrm{Recall}}{\mathrm{Precision}+\mathrm{Recall}})$$
where TP, TN, FP and FN are true positives, true negatives, false positives, and false negatives, respectively.

#### 4.3.2. Experimental Results

Within this subsection, we will experiment and discuss the neural network structure proposed in Section 3.2 with the parameters shown in Table 4. This paper uses 10-fold cross-validation to randomly divide the dataset into 10 subsets, with each remaining 9 subsets used for training, and 1 subset used for validation in 10 cycles.

UAV presence detection is actually a binary hypothesis problem, and for the detector (to detect the presence of UAV), the RF background activity without UAV signal and the RF signal with UAV are marked as zero and one, respectively, and then fed into the UAV detection network in Table 1 in Section 3.2. Using 10-fold cross-validation, 30 rounds are trained in batches of 10, and the last activation layer is chosen as a Sigmoid function as well as convergence using the MSE loss function. The final binary classification accuracy and F1 scores reach 100%, as shown in Figure 4a.

If the Detector identifies it as the UAV signal, we need to further classify its UAV type. The RF background activity signal without UAV signal, Bebop, AR, and Phantom are labeled as zero, one, two, and three, respectively; and finally, the data are fed into the CNN network in Table 2 in Section 3.2. By 10-fold cross-validation, 200 rounds are trained in batches of 20, and the last activation layer is chosen as a Softmax function as well as convergence using a Cross-Entropy loss function. The final quadruple classification accuracy was 99.6% and the F1 score was 99.6%, as shown in Figure 4b.

Finally, we further identify which mode the RF signal classified by the type classifier (Classifier 1) is in. For these two classifiers, we also use the CNN network structure from Table 2 in Section 3.2 for classification. The four patterns are also labeled and fed into the CNN network, which is trained for 300 rounds in batches of 20 using 10-fold cross-validation, with the last activation layer chosen as a Softmax function and a cross-entropy loss function for convergence. Finally, to better compare with other papers using this dataset, we took the background RF activity, the four types of AR/Bebop patterns, and the Phantom type, to form a 10-class confusion matrix with a classification accuracy of 99.3% and an F1 score of 99.3%, shown in Figure 4c.

Worth noting: the rows and columns of this matrix’s interior correspond to the predicted and true classes, respectively. The diagonal cells highlighted in green indicate correctly classified segments, while the non-diagonal cells indicate incorrectly classified segments, with the total number of segments and percentage of segments in each cell in bold. Next, the grey column on the far right shows Precision in green and FDR in red; the grey row at the bottom shows Recall in green and FNR in red; the blue cell in the bottom right corner of the figure shows Overall Precision in green and Error in red; the yellow column and row on the far left and top show the F1 score predicted for each class in green and it shows complementarity (completeness) in red; the orange cell in green indicates the average F1 score for all classes, and in red its complementarity.

## 5. Comparison with Other Methods

We compare the method proposed in this paper with other papers using the same dataset [27,28,29,30]. In paper [29], the authors used four fully connected layers to build a DNN network and utilized 10-fold cross-validation for training to verify the validity of the dataset; in paper [30], the authors tried to use CNN instead of DNN to solve the detection problem and obtain better accuracy; and in paper [27], the authors used only the lower band of RF signals and XGBoost algorithm to solve the classification problem. In the paper [28], the authors combine KNN and XGBoost to solve the pattern classification problem of UAVs using an integrated learning approach. Table 5 summarizes the performance of the five methods, and the results demonstrate that: firstly, the effectiveness of using compressed sensing techniques on sampling the communication signals between the UAV and the controller is verified; and secondly, it shows that the method used in this paper outperforms other learning methods in terms of accuracy, F1 score, and recall, at very low sampling rates.

## 6. Conclusions and Future Work

In this paper, we introduce the compressed sensing technique into the sampling of the session communication signal between the UAV and controller with a multi-channel random demodulator constructed. For the detection and identification of UAVs, we use the DNN and CNN networks designed in this paper for UAV detection and classification and finally achieve 100% accuracy for detecting UAV presence, 99.6% accuracy for identifying UAV types, and about 99.3% for UAV pattern classification. It is fully verified that the proposed signal sampling method and UAV detection algorithm in this paper is effective and can still obtain good detection results under a very low sampling rate. In future work, we intend to use multi-station detection to achieve UAV localization and tracking under the reduced pressure of data storage, transmission, and processing, through compressed sampling in this paper.

## Figures and Tables

**Figure 1 sensors-22-03072-f001:**
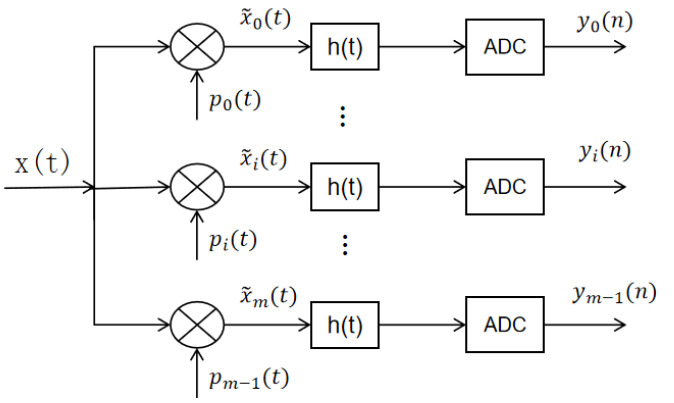
Multi-channel Random Demodulation Sampling Structure.

**Figure 2 sensors-22-03072-f002:**
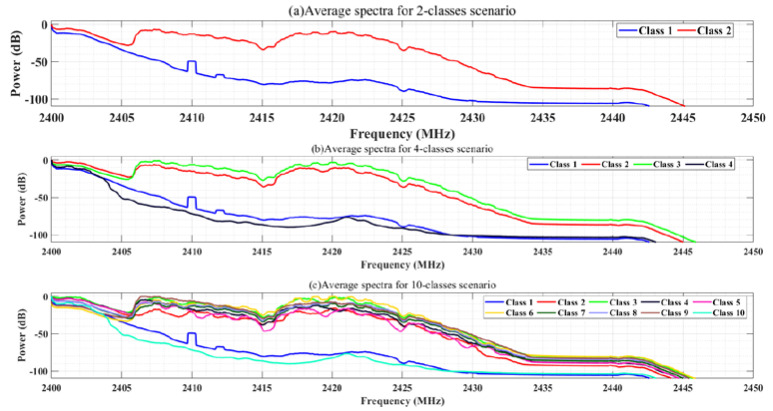
Average power spectrum of the RF activities for 2, 4, and 10 classes scenarios: In (**a**), class 1 is for RF background activities and class 2 is for the drones RF communications (to be supplied to the UAV Detection Network). In (**b**), class 1 is for RF background activities and classes 2–4 are for the Bebop, AR, and Phantom drones (to be supplied to the UAV Identification Network). In (**c**), class 1 is for RF background activities, classes 2–5 are for the Bebop 4 different flight modes, classes 6–9 are for the AR 4 different flight modes; and lastly, class 10 is for the Phantom single flight mode (to be supplied to the UAV Classification Network).

**Figure 3 sensors-22-03072-f003:**
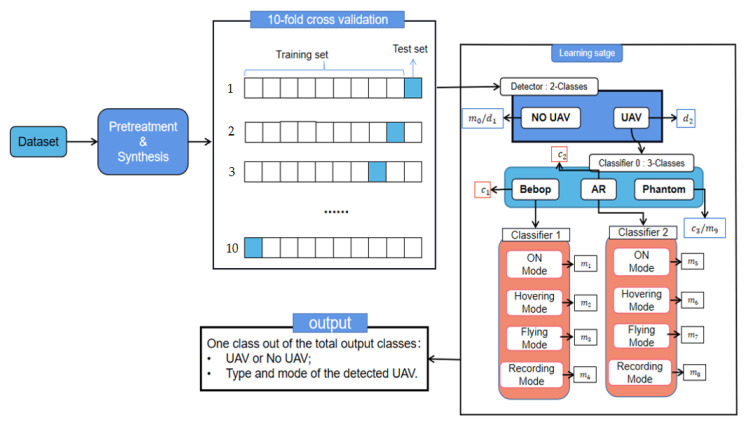
Our approach consists of dividing the dataset into training and testing data stage and a learning stage with four classifiers to specify the class of any sample.

**Figure 4 sensors-22-03072-f004:**
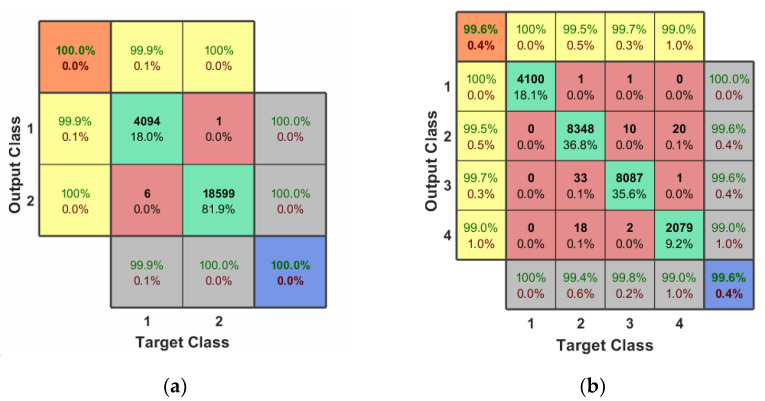
The output confusion matrix for 2-classes, 4-classes, and 10-classes scenario: Where (**a**) is the 2-classes output confusion matrix (detecting the presence of a drone); (**b**) is the 4-classes output confusion matrix (detecting the presence of a drone and identification its type); and (**c**) is confusion matrix of outputs for 10-classes scenario (detecting the presence of a drone, identification its type, and determining its flight mode).

**Table 3 sensors-22-03072-t003:** Composition of Drone-RF Dataset.

Class-2	Class-4	Class-10	Segments	Ratio
No UAV	No UAV	No UAV	41	18.06%
UAV	Bebop	On, connected, off	21	9.25%
		Hovering	21	9.25%
		Flying	21	9.25%
		Flying with video recording	21	9.25%
	AR	On, connected, off	21	9.25%
		Hovering	21	9.25%
		Flying	21	9.25%
		Flying with video recording	18	7.94%
	Phantom 3	On, connected, off	21	9.25%

**Table 4 sensors-22-03072-t004:** Selection of Experimental Parameters.

Experiments	Epochs	Batch Size	Activation Functions	Loss	Learning_Rate
Detector	30	10	Sigmoid	mse	Default
Classifier1	200	20	Softmax	CrossEntropy	Default
Classifier23	300	120	Softmax	CrossEntropy	0.00003

**Table 5 sensors-22-03072-t005:** Results of different sampling methods and machine learning.

Reference	Sampling	Approach	2-Classes	10-Classes
[29]	NS ^1^	DNN	99.7%	46.80%
[30]	NS ^1^	CNN	99.8%	59.20%
[27]	NS ^1^	XGBoost	99.9%	70.09%
[28]	NS ^1^	Hierarchical	99.5%	99.20%
Ours	CS ^2^	DNN/CNN	100%	99.30%

^1^ is sampling theory, ^2^ is Compressive Sensing.

## Data Availability

The data used in this study are available online under this website https://data.mendeley.com/datasets/f4c2b4n755/1 (accessed on 1 September 2021).

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
