# Peer review of "Deep Learning Approach to UAV Detection and Classification by Using Compressively Sensed RF Signal"

_sensors, 2022, doi:10.3390/s22083072_

Round 1

Reviewer 1 Report

The authors propose in this paper an algorithm for UAV detection and classification by using compressed sensing and deep learning techniques. The subject is of high interest nowadays, due to increasing number of incidents generated by UAVs entering some protected areas, accidentally or intentionally. The authors’ arguments for selecting compressed sensing and deep learning techniques are the highly intermittent RF signal for the UAV control and the large amount of data to be processed, respectively. A review of representative papers using one or both of the techniques is presented and some significant results are enumerated.

Next, the specific implementation of the compressed sensing techniques in their proposed algorithm is analytically described, underlying the benefits brought by their approach of splitting the RF bandwidth in smaller units, applying compressed sensing on each unit and, finally, concatenating all the results to obtain the total power spectrum. The resulting data are successively fetched to two specifically designed neural networks (NN) for the UAV detection and its type and operation mode classification, respectively. The authors present in detail the architectures of the implemented NNs and the values of the parameters used for experimentation.

In order to validate their algorithm, the authors use a public available data set containing information about three types of UAVs and four operation modes. An initial statistical analysis of the selected data set shows that the average power spectra clearly depend on the UAV type and on its operation mode, but the difference is too small to be revealed by means of traditional signal recognition techniques, so using dedicated NNs is highly recommended.

The authors conducted a great number of experiments and their results are synthetically presented by means of usual confusion matrices. They report a surprisingly 100% success in detecting the presence of an UAV and a 99.3% success rate in classifying its UAV type and operation mode. Their results are superior to the ones found in the literature and comparatively presented in a table. The authors intend to generalize their algorithm for detecting and classifying multiple UAVs.

The paper is well structured, the literature review on the subject is comprehensive, the proposed algorithm is clearly explained, and the results are properly displayed and commented.

The following changes in the paper presentation could bring more clarity and consistency:

  • a more careful description of data pre-processing analytics: rel. (6) – limits of variable k, rel. (8) – nominator expression, meaning of operator * and of variable (q)’ in the denominator, specific criteria for choosing the value of Q;
  • complete citation of paper [24], used as reference for the proposed algorithm.

An interested reader would have difficulties in following the paper ideas, due to some clumsiness in formulating the sentences. A language review by an expert is highly desirable.   

Reviewer 2 Report

In this paper, a DNN and CNN for UAVs detection and classification method is proposed, where the RF communication signal is sampled using the compressed sensing technique. Simulation and real dataset are provided to validate the advantage of the proposed algorithm. I have the following suggestions:

1) Lots of cross references is missing, please correct.

2) The scheme diagram of your proposed UAV detection and classification method should be provided. Also, the network architecture used in this paper should also be presented.

3) P2, line 58, “…UAV detection algorithms based on acoustics [6,7], visual (image or video) [8,9], radar [10],…”, here, more radar detection methods should be added, such as [1-3].

[1] "Target Detection Within Nonhomogeneous Clutter Via Total Bregman Divergence-Based Matrix Information Geometry Detectors," in IEEE Transactions on Signal Processing, vol. 69, pp. 4326-4340, 2021, doi: 10.1109/TSP.2021.3095725.

[2] "Persymmetric Subspace Detectors With Multiple Observations in Homogeneous Environments," in IEEE Transactions on Aerospace and Electronic Systems, vol. 56, no. 4, pp. 3276-3284, 2020.

[3] "Detection of the Number of Signals by Signal Subspace Matching," in IEEE Transactions on Signal Processing, vol. 69, pp. 973-985, 2021, doi: 10.1109/TSP.2021.3053495.

4) In Section 2, when you introduce the classical UAV detection method, lots of signal detection methods, e.g., [1-3] (mentioned above), should be added, since these methods can be used to detect UAV directly by resort to the communication signal or radar signal, ect.

5) You should give the reason why you choose the DNN and CNN to detect and classify the UAV but not other neural networks.

Round 2

Reviewer 2 Report

The authors have addressed all my concerns. In addition, the missing authors information in the references should be corrected.

Author Response

Dear Reviewer,

Hello! Thank you for your acknowledgement of this revision and the corresponding reminder of your suggestions. Below is the addition of the corresponding information.

Point 1: The missing authors information in the references should be corrected..

Response 1: I have added the relevant author information in the references. Details are as follows.

Yongguang Mo received his Bachelor's degree in Electronic and Information Engineering from the School of Information Engineering, Jiangxi University of Science and Technology, China in 2019, and he is now a postgraduate student at Shenzhen University. His research interests include intelligent information processing, detection and tracking of drones.

Jianjun Huang (M’02) received the Ph.D degree in Signal and Information Processing from Xidian University in 1997. He was a Postdoc in Computer Sciences with Northwestern Polytechnical University from 1997 to 1999. He is now a professor with College of Electronics and Information Engineering, Shenzhen University, China. His current research interests include intelligent information processing, computer vision and medical image processing.

Gongbin Qian received the B.Eng. and M.Sc.degrees from the Harbin Institute of Technology,Harbin, China, in 1990 and 1993, respectively, and the Ph.D. degree from the South China Universityof Technology, Guangzhou, China, in 2009. He is currently an Associate Professor with Shenzhen University.His research areas include communication theory, signal processing for communications, and array signal processin.